# Soil Erosion and Gaseous Emissions

**Rattan Lal**

Carbon Management and Sequestration Center, the Ohio State University, Columbus, OH 43210, USA; lal.1@osu.edu

**Abstract:** Accelerated soil erosion by water and wind involves preferential removal of the light soil organic carbon (SOC) fraction along with the finer clay and silt particles. Thus, the SOC enrichment ratio in sediments, compared with that of the soil surface, may range from 1 to 12 for water and 1 to 41 for wind-blown dust. The latter may contain a high SOC concentration of 15% to 20% by weight. The global magnitude of SOC erosion may be 1.3 Pg C/yr. by water and 1.0 Pg C/yr. by wind erosion. However, risks of SOC erosion have been exacerbated by the expansion and intensification of agroecosystems. Such a large magnitude of annual SOC erosion by water and wind has severe adverse impacts on soil quality and functionality, and emission of multiple greenhouse gases (GHGs) such as $CO_2$, $CH_4$, and $N_2O$ into the atmosphere. SOC erosion by water and wind also has a strong impact on the global C budget (GCB). Despite the large and growing magnitude of global SOC erosion, its fate is neither adequately known nor properly understood. Only a few studies conducted have quantified the partitioning of SOC erosion by water into three components: (1) redistribution over land, (2) deposition in channels, and (3) transportation/burial under the ocean. Of the total SOC erosion by water, 40%–50% may be redistributed over the land, 20%–30% deposited in channels, and 5%–15% carried into the oceans. Even fewer studies have monitored or modeled emissions of multiple GHGs from these three locations. The cumulative gaseous emissions may decrease at the eroding site because of the depletion of its SOC stock but increase at the depositional site because of enrichment of SOC amount and the labile fraction. The SOC erosion by water and wind exacerbates climate change, decreases net primary productivity (NPP) and use efficiency of inputs, and reduces soils C sink capacity to mitigate global warming. Yet research information on global emissions of $CH_4$ and $N_2O$ at different landscape positions is not available. Further, the GCB is incomplete and uncertain because SOC erosion is not accounted for. Multi-disciplinary and watershed-scale research is needed globally to measure and model the magnitude of SOC erosion by water and wind, multiple gaseous emissions at different landscape positions, and the attendant changes in NPP.

**Keywords:** global carbon budget; soil organic carbon erosion; deposition; gaseous emissions; enrichment ratio; soil depletion; preferential removal

## 1. Introduction

As a natural geological process, soil erosion over eons has created the world's most fertile alluvial and aeolian (loess) soils. Acceleration of the natural erosion process by human activities, ever since the dawn of settled agriculture ~12 millennia ago, has caused the most severe environmental problems of the 21st century. Soil erosion, involving breakdown and transport of soil particles, requires energy, and a specific type of erosion depends on the source of energy (Figure 1). Water and wind are among the principal sources of energy, and thus major factors of erosion. Being a selective process, soil erosion removes and transports fine (i.e., clay and silt) and light (soil organic carbon or SOC) fractions. These three constituents (i.e., clay, silt, and SOC) are also key determinants of soil quality, and its capacity to provide numerous ecosystem services (ESs). However, these essential constituents



are depleted over time in soils prone to accelerated erosion. The latter has plagued the Earth and humanity for millennia. The data based on the analysis of sediments from 600 lakes worldwide show that anthropogenic activities accelerated global soil erosion 4000 years ago [1]. Many once-thriving civilizations vanished because they treated their soil like dirt [2,3]. The current problem of accelerated soil erosion is driven by a rapid and an indiscriminate expansion of agroecosystems for feeding the growing population. Further, the problem of soil erosion is also exacerbated by anthropogenic global warming [4,5]. In addition to adversely impacting the wellbeing of 3.2 billion people [6], accelerated soil erosion is also polluting the environment (i.e., soil, water, and air). It affects and is affected by the present and will be aggravated by the projected climate change.

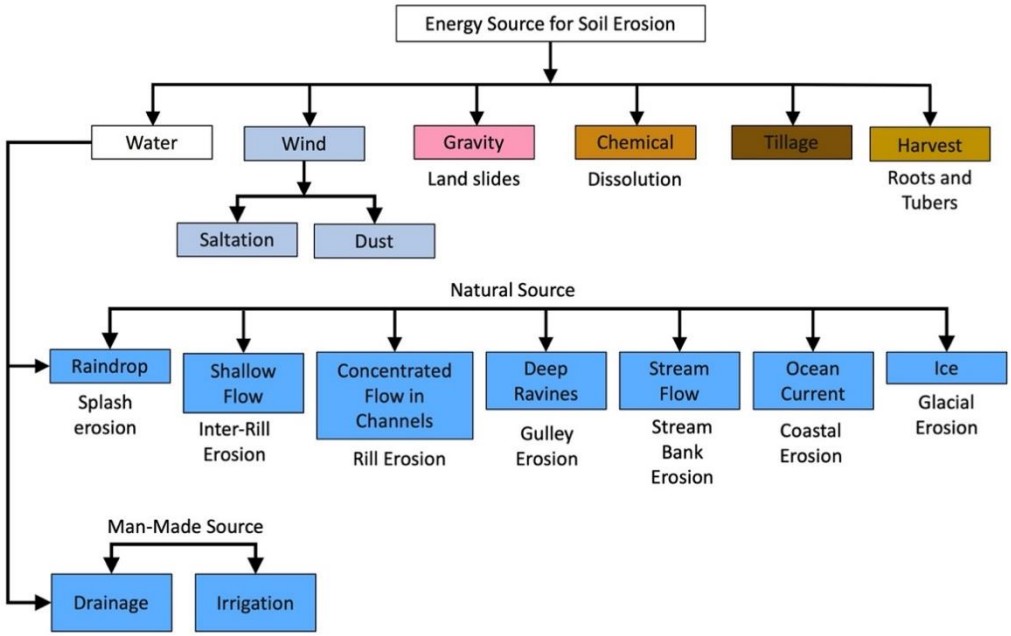

**Figure 1.** Types of soil erosion driven by source of energy.

Under natural ecosystems, SOC stock is a sink of atmospheric carbon dioxide ($CO_2$), and is protected against microbial processes through the formation of organo-mineral complexes and stable structural units or aggregates. Conversion of natural to managed ecosystems disrupts aggregates, exposes the hitherto protected SOC, and increases its vulnerability to transport by erosion and decomposition by microbial processes. Preferentially removed light SOC fraction is redistributed over the landscape, deposited in channels and transported to aquatic ecosystems and depressional sites (Figure 2). The labile SOC fraction is exposed to microbial processes when being transported, and following after redistribution and deposition phases of the erosion process. Furthermore, the historic land use based on extractive farming practices also mined off the SOC stock as a source of plant nutrients. Thus, soils of most agroecosystems are depleted of their original SOC stock. Consequently, soil quality is degraded, the capacity to perform ESs is impaired, and the environment (i.e., soil, water, air, and biodiversity) jeopardized.

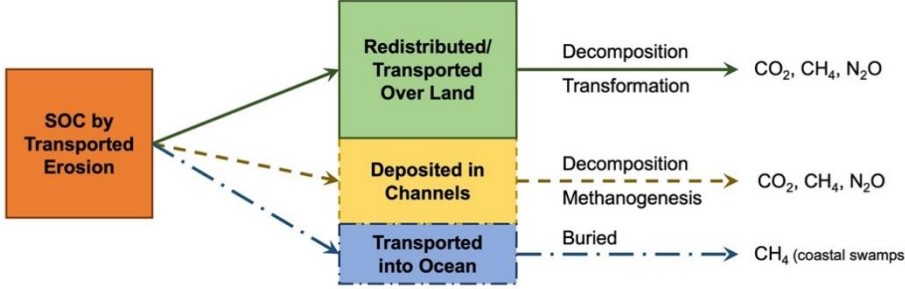

**Figure 2.** The fate of soil organic carbon transport by erosion. About 40%–50% may be redistributed over the land, 20%–30% may be deposited in channel, 5%–15% may be carried into the ocean, and about 15%–20% may be emitted into the atmosphere. However, the exact partition may vary among soil, climate, land use, and other site-specific factors. Whereas the cumulative emission of $CO_2$ may decrease at the eroded site, it may increase at the transported and depositional zones.

The global magnitude of historic depletion of SOC by all processes may be as much as 135 Pg C [7]. Consequently, degraded and depleted soils also have a large carbon (C) sink capacity to reabsorb atmospheric $CO_2$ into SOC stock upon conversion to a restorative land use and adoption of conservation-effective practices.

It is this potential of restoring the global SOC stock, for advancing food and climate security and strengthening soils' capacity to provide ESs, that sustainable soil management is receiving the attention of policymakers. Ever since the launch of the 4 Per Thousand (4P1000) initiative at COP 21 in Paris in 2015 [8], world soils have been on the global agenda as an option to sequester C and mitigate global warming. Such initiatives are aimed at achieving greenhouse gas (GHG) neutrality through low-carbon farming [9].

Transport of C by accelerated soil erosion at a global scale is one such process that impacts the emission of $CO_2$, methane ($CH_4$), and nitrous oxide ($N_2O$). The drastic increase in SOC erosion by anthropogenic activities poses a daunting challenge of assessing its impact on the global C budget (GCB) and GHG emissions. Therefore, it is important to credibly assess the mean annual flux of GHGs from soils during different erosional phases so that the magnitude of the carbon dioxide equivalent ($CO_2$ eq) can be estimated. Whereas the soil C transported by erosional processes comprises of SOC and soil inorganic C (SIC), the fate of SOC transported by water and wind erosion that impacts the emission of GHGs [10] is not understood. Therefore, the objectives of this article are to describe the effects of erosion on the emission of GHGs into the atmosphere, explain processes affecting gaseous emissions by soil erosion, describe generic options that can reduce risks of soil erosion and minimize the emission of GHGs, and identify researchable priorities. This article is based on the hypothesis that accelerated soil erosion is a source of major GHGs including $CO_2$, $CH_4$, and $N_2O$ during all three phases of the erosional process.

## 2. Materials and Methods

The literature is replete with articles on soil erosion by water and wind. Thus, the literature search was specifically focused on available information on the magnitude of SOC transported by water and wind erosion was collated from the Web of Science, Google, and other sources. The literature search involved journals dealing with basic and applied sciences. The focus included journals dealing with: (a) earth sciences such as Global Change Biology, Global Biogeochemical Cycles, Biogeosciences, Geomorphology, J. Geophysical Research, Earth Surface Processes and Landforms, Geochemistry, J. Geophysical Res., J. Hydrology, Aeolian Research, (b) popular journals such as Science, Nature, Philosophical Transactions of Royal Society, (c) environmental sciences including Env. International, Climatic Change, Ecosphere, (d) journals devoted to soil science including Soil Research, Geoderma, Soil Sci. Soc. Amer. J., Catena, European J. Soil Sci., Australian J. Soil Res., J. Soil and Water

Conservation, and (e) those dealing with policy issues such as Land Use Policy, Science Policy, and Land Degradation and Development. Only those articles were selected for discussions in the present review which contained quantitative data on the magnitude of SOC or total carbon (TC) transported by erosional processes, and information on gaseous emissions at different landscape positions within an eroding landscape. While the literature searched is global, most of the articles addressing this theme were those published from the research done in the U.S.A., Europe, East Asia, Australia, and South America.

*2.1. Soil Erosion by Water: Transport, Redistribution, and Deposition of Soil Organic Carbon Over the Landscape*

Water erosion affects as much as 1.1 billion hectares (B ha) of the land area [11]. Available data on the magnitude of sediment load transported by world rivers are more credible [12] than that for the amount of soil moved by aeolian processes. The global land–ocean flux of sediment has reportedly increased from 14.0 Pg/yr. (Pg = peta gram = 1 billion metric ton) during the pre-human era to the contemporary flux in the absence of reservoir trapping to 36.6 Pg/yr. [12]. Sediments are enriched in SOC, and the global increase in sediment load may cause a strong increase in the transport of SOC, whose fate must be understood in relation to emissions of GHGs. Soil erosion on U.S. cropland increased by ~17% over the 20th century through the expansion of the land area under agriculture [13]. The SOC fraction entrained in the shallow runoff is moved and redistributed over the landscape. Erosion of soil and SOC stock has direct and indirect effects on soil and environment quality, net primary productivity (NPP), and efforts to achieve land degradation neutrality or LDN (Figure 3). The magnitude of the effect of emissions of GHGs is governed by the pathways of SOC erosion. The fate of SOC being redistributed depends on how it is being moved by the fluvial processes and on the temperature and moisture regimes at the redistribution and depositional positions (Figure 2). Quantitative assessment of the movement of SOC over the landscape is essential to establishing the watershed level C budget [14] that can be scaled up to the river basin and eventually to regional, national, or global scale. The magnitude of SOC erosion by fluvial processes varies widely (Table 1) depending on a range of factors. Important among these are climate [10,13], soil [15–17], terrain [18] and land use [13–15,19–23]. On the basis of some empirical data from 240 runoff plots studied over the entire rainy season from diverse global ecoregions, Mueller-Nedebock and Chaplot [18] estimated that the total amount of SOC displaced by sheet erosion from its source would be 1.32 ± 0.20 Pg C, or about 11.4% of the annual anthropogenic emission of 11.5 Pg C in 2019 [21]. Integrating all C fluxes for the EU agricultural soils, Lugato et al. The author of [24] estimated a net C loss or gain of −2.28 Tg $CO_2$ e/yr. and +0.79 Tg $CO_2$ e/yr., and they argued that strong agricultural policies are needed to prevent or reduce soil erosion. Assessing and accounting for all the additional feedback and C fluxes due to displacement by erosion, Lugato et al. [15] estimated a net source of 0.92 to 10.0 Tg C/yr. from agricultural soils in the European Union to the atmosphere over the period of 2016–2100.

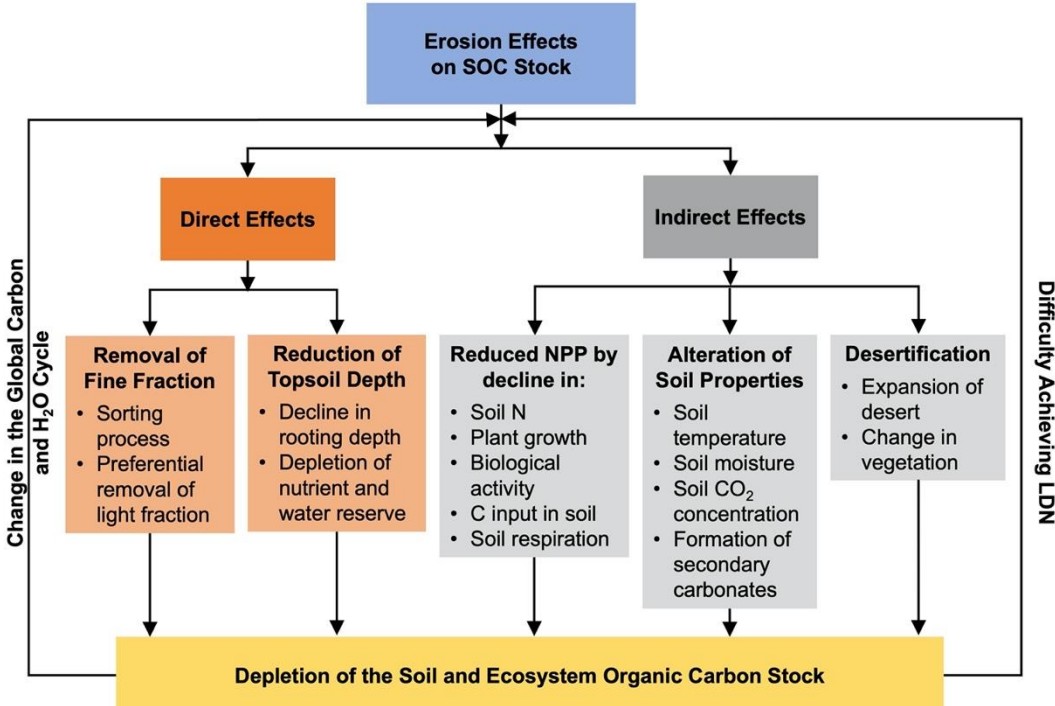

**Figure 3.** Effects of accelerated soil erosion on the global carbon cycle and the increase in the daunting challenge of achieving land degradation neutrality.

**Table 1.** Examples of regional, national, or global terrestrial soil organic carbon (SOC) erosion by water and other processes.

| Country/Region | Study Duration (yr) | SOC Erosion | Erosion Types | References |
|---|---|---|---|---|
| Australia | 40 | 4 Tg SOC/yr. | all processes | [25] |
| Burkina Faso | — | 0.15–0.37 g C/m$^2$·yr. | Water | [16] |
| China | 20 | 180 ± 80 Mg C/yr. | Water | [26] |
| European Union | — | 0.05–0.45 Mg C/ha·yr. | Water | [24] |
| Global | — | 1.32 ± 2 Gt C/yr. by sheet erosion | Water | [18] |
| Global | — | 1.1 Pg C/yr. flux | Water | [10] |
| Global | 150 | 0.49 ± 0.12 Pg C/yr. | Water | [27] |
| India | — | 115.4 Tg C/yr. | Water | [28] |
| Spain | — | 0.031 ± 0.03 Mg C/ha·yr. | Water | [14] |
| Turkey (Seyhan River Basin) | — | 0.19 Mg C/ha·yr. | Water | [29] |

The SOC being eroded is either deposited in the landscape, in the channel, or carried into the ocean (Figure 2). Some of the SOC being transported is emitted into the atmosphere as $CO_2$ or $CH_4$, depending on the degree of wetness or anaerobiosis. In China, Fang et al. [17] observed that 42% of the eroded SOC was redeposited within the catchment. The mean residence time (MRT) of the deposited C depends on a range of site-specific factors, and the fraction composition (labile, intermediate, passive) of the eroded SOC. Wang et al. [30] reported that cumulative emission of soil $CO_2$ decreased slightly at the erosion site but increased by 56% and 27% at the transport and depositional zones, respectively, in comparison to non-eroded sites. Wang and colleagues concluded that overall, $CO_2$ emissions contributed 90.5% of the total erosion-induced C loss over the 4-month experiment. Whereas buried SOC at depositional sites may have a higher MRT even for the fast and intermediate turnover pools [31],

susceptibility to decomposition may be much higher for the labile fractions redistributed within the landscape.

Examples of the magnitude of SOC erosion by water from different regions are shown in Table 1 and may range from 1.1 to1.3 Pg C per year. The preferential removal of SOC by water erosion is indicated by a high enrichment ratio of SOC (and clay) in sediments compared to that of the surface soil from which the sediments originated. Consequently, the enrichment ratio for SOC in alluvial sediments is >1 and may be as high as 12 (Table 2). Erosional processes lead to a preferential transport of SOC because it has a low bulk density and is concentrated in the surface soil layer. In cases where sediments are derived from subsoil (i.e., gully erosion), the enrichment ratio can be less than 1 [32].

There are a few studies involving techniques of quantitative measurement of SOC/TC transported by erosion from a watershed or a well-demarcated area. In Australia, Chappell et al. [25] estimated the magnitude of $^{137}$Cs -derived redistribution of SOC by all processes (water, wind, and tillage) at 4 Tg SOC/yr for 1950–1990s. This represents an average loss of 2% of TC stock, assuming that total C is mineralized as $CO_2$, and would represent a net national flux of 15 Tg $CO_2$ eq/yr from all C pools in Australia. In a follow-up study, Chappell et al. [33] estimated the global terrestrial SOC erosion at 0.3–1.0 Pg C/yr. For the Seyhan River Basin in the Mediterranean region of Turkey, the rate of SOC erosion was estimated at ~0.2 Mg C/ha·yr [20].

**Table 2.** The enrichment ratio of carbon in sediments derived from water erosion.

| Country | Experiment | Enrichment Ratio | Reference |
|---------|------------|------------------|-----------|
| Australia | Rainfall simulation | >1 | [34] |
| Belgium | Rainfall simulation | 1.2–3.0 | [35] |
| Belgium | Rainfall simulation | 0.9–2.6 | [36] |
| China | Lab studies | 1.3–4.0 | [37] |
| China | Rainfall simulation | 0.98–1.01 | [32] |
| USA | Field experiment under cotton | 8–12 | [38] |

*2.2. Wind Erosion*

Wind erosion, affecting about 550 million hectares of the global land area [11,39], is caused by aeolian (or eolian) processes. The term "aeolian" is derived from the Greek god "Aeolus", the keeper of the wind. Wind erosion is strongly affected by soil texture. Soils most susceptible to wind erosion may have <5% clay and <3% silt, and >50 cm deep surface layer [22]. Wind erosion may create 500–5000 Tg (million tons) of dust annually with a strong impact on soil properties, air quality, and human health [39–41]. The environmental impacts of wind erosion during the Dust Bowl Era of 1931 through 1939 are described by Steinbeck [42].

Accelerated erosion affects critical biotic and abiotic processes governing the soil/ecosystem C cycle. The magnitude of the loss of SOC by wind erosion is related to that of the fine soil fraction [43]. The loss of C-enriched fine soil particles depletes its SOC and reduces its future potential to restore the SOC pool. The aeolian erosion process affects both progressive and regressive pedogenesis in dry eco-regions. On agricultural lands, erosion degrades soil quality by removal of silt, clay, and SOC fractions through effective sorting processes that leave behind only coarse sand and gravels [23]. The loss of NPP reduces the plant feedback and aggravates the SOC loss [44].

The wind-blown dust is also enriched in SOC, which may also depend on soil texture. The global estimate of SOC erosion by wind may be as much as 0.3 to 1.0 Pg C/Yr (Table 3) and some highly vulnerable soils may lose 3.6 Mg C/ha per year [32]. Losses of SOC by wind erosion in Northern China are estimated at 0.9 Tg C/yr [45]. The loss of PM10 (particles of <10 micrometer) adversely impacts soil nutrient reserves [40]. Wind-blown dust is also enriched in SOC and has a high enrichment ratio (Table 4). In Niger, Sterk et al. [46] assessed nutrient and C losses in saltation and suspension transport by conducting chemical analysis of the trapped material at 0.05, 0.26, 0.5, and 2 m. The sediments were three times richer than topsoil at 0.5 m and 17 times at 2 m. In Australia, Webb et al. [47] observed that

the SOC-enrichment ratio ranged from 2.1–41.9 for a sandy and 2.1 for clayey soil (Table 4). Webb and colleagues hypothesized that in addition to particle size, distribution, and the degree of aggregation, size-selective sorting of SOC during transport may enhance the enrichment of SOC dust emissions. The SOC concentration in two of the dust samples was 15%–20% by weight. A study in China by Ravi et al. [48] documented that an increase in the particulate matter emissions (e.g., black earth) from biochar-amended soils may counteract the negative emissions potential of biochar. Magnitude of dust emitted is aggravated by human activities [40].

**Table 3.** Examples of regional, national, or global terrestrial SOC erosion by wind.

| Country/Region | Study Duration (yr) | SOC Erosion | Erosion Types | References |
|---|---|---|---|---|
| Australia | — | 3.6 Mg C/ha | wind | [22] |
| China | 1–3 | 34–39 Tg C/yr. | wind | [30] |
| China | 10 (1990s) | 75 Tg C over 10 yr. | wind | [49] |
| China | 56 (1954–2010) | 92.8 kg C/ha·yr. | wind | [17] |
| China (NW China) | 34 (1980–2013) | 27.5 Tg C | wind | [44] |
| Global | 100 | 0.3–1.0 Pg C/yr. | wind | [50] |
| Hungary | 10 min | 2.25–2.50 g C/m$^2$ | wind | [51] |

**Table 4.** The enrichment ratio of carbon in sediments derived from wind erosion.

| Country | Experiment | Enrichment Ratio | Reference |
|---|---|---|---|
| Australia | Field experiments | 1.7–7.1 | [47] |
| Australia | Field sites | 2.1–41.9 sand-rich soil <br> 2.1 clay soil | [52] |
| Canada | Field sites | 1.05 | [53] |

Both direct and indirect effects of accelerated erosion, especially in arid and semi-arid regions, exacerbate the risks of desertification and drastically increase the already daunting challenge of achieving land degradation neutrality or LDN by 2030 [54,55]. Soil degradation impacts of accelerated erosion by wind, and its positive feedback to the process of desertification, have strong adverse consequences on Earth systems and human environments [40], as well as on NPP, the input of biomass-C into the soil, and on the disruption of the global C cycle. Thus, achieving LDN would necessitate the global adoption of conservation-effective measures that reduce risks of both aeolian and alluvial processes of soil erosion [56]. Soil restoration strategies must be directed towards increasing the input of biomass-C into the soil. Increase in NPP, and the attendant increase in the input of biomass-C into soil, would restore SOC stock [57]. Dust emission caused by wind erosion may be aggravated by the projected climate change. Thus, Duniway et al. [58] recommended multidisciplinary and multijurisdictional approaches and perspectives to understand the complex processes of dust emission and identify strategies of its mitigation.

*2.3. Gaseous Emissions from Eroded Sediments and the Fate of Carbon Transported and Deposited over the Landscape*

Soil C stock is an important component of the global C cycle. The historic C loss from soil may have emitted as much as 537 Pg C or 27% of the amount present before the onset of agriculture about 10 millennia ago [59]. Erosion and redistribution disturb a large quantity of soil C in managed and natural landscapes. The fate of soil C impacted by erosion may differ among the sites of erosion, redistribution, and deposition (Figure 2). Some examples of gaseous emissions from eroded and depositional sites are shown in (Table 5) [60,61]. Assuming an average flux of 300 mg $CO_2$ eq/m$^2$·h based on literature review, Oertel et al. [62] estimated the global annual net soil emissions at ≥350 Pg $CO_2$ e, as compared with the 2018 anthropogenic emission of 42.1 Pg $CO_2$ e [21]. However, the large emissions from soil C transported by aeolian and alluvial processes are not considered in the global C budget

(GCB), which creates a lot of uncertainty. Nonetheless, understanding, managing, and reducing the erosion-induced gaseous flux of $CO_2$, $CH_4$, and $N_2O$ (Figure 2) is an important researchable priority to reduce uncertainty in the GCB. It is also critical to identify hot spots (vulnerability, resilience, and action), and plans of targeted interventions for managing the flux [63]. Several pedological processes impacted by erosion also affect NPP through alterations in availability and uptake of water, nutrients, and photosynthesis. Credible assessment of C dynamics in agricultural and other landscapes is important to addressing global issues [19]. In a Mediterranean Seyhan river basin, Cilek [29] estimated SOC erosion of 0.163 Mg C/ha yr. (total SOC loss of 349, 850 Mg C/yr. over a total watershed area of 21,485 km$^2$). Based on the assessment of nine river basins in China, Wang et al. [30] found that total SOC erosion was 68.4 and 77.3 Tg C/yr. for 1995–1996 and 2010–2012, respectively. Of this, 57% and 47% were redistributed over land, 25% and 44% was deposited in channels, and 18% and 8% were delivered into the sea, respectively. However, how much and which gases were emitted were not determined. For the period A.D. 1850–2005, Naipal et al. [27] estimated global SOC flux of 47 ± 18 Pg C, of which 79%–85% occurs on agricultural and grasslands.

**Table 5.** Examples of gaseous emission from eroded sediments and disrupted/broken aggregates by erosional processes.

| Country | Experiment | Emission | Reference |
| --- | --- | --- | --- |
| U.K. | Field | Emission factor of 5.5, 4.4., and 0.3 Mg $CO_2$ Eq/yr*Mg of fluvial C, gross C erosion, and gross soil erosion, respectively | [60] |
| South Africa | Simulated Rain | 0.031–0.039 gC $CO_2$/g C | [61] |

*2.4. Implications of Ignoring Erosion Induced Transport of Carbon in Estimating Global Carbon Budget*

Erosion-induced transport of soil C (SOC and SIC) is a large and growing component with a strong impact on the GCB, but which is now being omitted [21]. However, the erosion-induced impact on soil C stock and flux, a large component comprising of multiple gases ($CO_2$, $CH_4$, $N_2O$) and multiple processes (e.g., water, wind, gravity, and tillage), must be dually considered. High-resolution models [20] must be developed to improve methodological protocols to account for this serious omission. By credibly accounting for the effects of erosion on net C exchange between the soil and the atmosphere, it may be possible to identify global hot spots of undertaking targeted interventions to mitigate the erosion-induced positive feedback to global warming. In Australia, Chappell et al. [25] found SOC erosion by all processes at 4 Tg/yr. (or 2% of total C stock in 10-cm depth). Assuming that most of this is mineralized, Chappell and colleagues estimated a flux of ~15 Tg $CO_2$ e/yr. representing 12% emissions from all C pools in Australia and concluded that it was an important source of uncertainty in the national carbon budget. By extending this study globally, Chappell et al. [50] estimated global terrestrial SOC erosion of 0.3–1.0 Pg C/yr, highlighted the significance of ignoring it in the accounting of the GCB, and suggested that accounting for SOC erosion would reduce uncertainty in the GCB.

**3. Conservation: Effective Measures for Reducing SOC Erosion and Afforestation of Eroded Lands for Sequestration of Atmospheric $CO_2$ in the Terrestrial Biosphere**

During the 1950s to 1990s, the objective of erosion control was to conserve soil and water, reduce the loss of soil fertility, and minimize the risks of non-point source pollution. Since the 1990s, two among important objectives of soil conservation and effective erosion control measures are to: (1) promote low-C agriculture (9), reduce risks of water runoff and soil erosion so that the attendant emission of GHGs can also be reduced from SOC erosion, and (2) to sequester atmospheric $CO_2$ in soil and vegetation through the restoration of eroded soils and desertified ecosystems. Examples of technological options to accomplish these include the adoption of conservation agriculture including no-till farming with retention of crop residue mulch [4,64–67] and use of cover cropping during the off-season [68] that reduce the risks of water runoff [69], boost SOC stock for food and climate [8,67], and

curtail transport of SOC and nutrient-enriched sediments [70] for accomplishing objective 1. Similarly, promoting afforestation [71] and adopting the concept of "Reducing Emissions from Deforestation and Forest Degradation or REDD [72] through afforestation of degraded soils [73], and establishments of shelterbelts in areas prone to wind erosion [74–76] would be pertinent to advancing objective 2 of sequestration of atmospheric $CO_2$ in the terrestrial biosphere.

## 4. Conclusions

The synthesis and a critical review of the literature presented above indicate that the hypothesis of the study is proven. Over and above the offsite effect of sedimentation and non-point source pollution, accelerated soil erosion is also an important source of major GHGs ($CO_2$, $CH_4$, and $N_2O$) emitted during all three stages of the erosion process.

The discussion presented has also led to accomplishing the major objectives of the study:

1.  Accelerated soil erosion (i.e., water, wind, and other processes) has a strong impact on the GCB. Soil carbon transported by erosional processes is partitioned as follows: (i) redistribution over the landscape, (ii) deposition in channels and other depressional sites, and (iii) transportation into the oceans. Because the eroded site is depleted of its SOC stock, its NPP is adversely affected, and the gaseous flux from soil to the atmosphere is decreased. Depending on the soil temperature and moisture regimes, gaseous fluxes (e.g., $CO_2$, $CH_4$, and $N_2O$) from redistribution and depositional sites may increase.

2.  The magnitude of soil C transported by water erosion may be 10%–15% of the total anthropogenic emissions of ~11.5 Pg in 2019. In addition, there is a large transport by wind erosion in arid and semi-arid climates. Such a large flux, with severe negative impacts on NPP and other biotic and abiotic processes, must be accounted for in the GCB. Considering the multi-gas flux ($CO_2$, $CH_4$, and $N_2O$), and the fact that some gases have a large global warming potential ($CH_4$ and $N_2O$), the magnitude of erosion-induced flux of GHGs must be accounted for and included along with those of other anthropogenic activities affecting global warming.

3.  The SOC erosion by water and wind accelerates anthropogenic climate change and also decreases soils' C sink capacity to mitigate global warming. Thus, accelerated soil erosion must be effectively managed to minimize the risks. Eroded soils, which are already severely depleted of their SOC stock, must be restored to improve soil health and enhance essential ecosystem services. Adoption of conservation-effective measures for restoration of eroded soils may represent a net significant carbon sink.

4.  Additional research is needed on developing methodologies to account for the SOC erosion into the GCB.

5.  Because SOC erosion is a source of multiple GHGs, erosion-induced emissions as $CO_2$ eq must be considered among the sources of anthropogenic emissions.

6.  Adoption of site/region-specific conservation-effective technologies (conservation agriculture, mulch farming, cover cropping, afforestation, REDD, and shelterbelts) must be promoted to conserve soil and water and reduce the emission of erosion-induced GHGs.

**Funding:** This research received no external funding.

**Acknowledgments:** Support received from Maggie Willis and Kyle Sklenka, the staff of the Carbon Management and Sequestration Center of the Ohio State University, is appreciated and acknowledged.

**Conflicts of Interest:** The authors declare no conflict of interest.

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
