# Peer review of "Soil Erosion and Gaseous Emissions"

_applsci, doi:10.3390/app10082784_

Round 1

Reviewer 1 Report

Dear author,

thank you for your study dealing with soil erosion and gaseous emissions. Your systematic study provides significant information about the impact of accelerated soil erosion on the Global C Budget (GCB). However, there are some aspects related to the aim and the layout of the manuscript not discussed and/or presented properly.

First, I would advise you to use the Microsoft Word template to re-organize the manuscript layout, because in the attached .pdf file there are some aspects that need improvement:

  • References in the text should be numbered in order of appearance and indicated by a numeral or numerals in square brackets. Please check the paper from line 123 and in Table 1
  • There are not the Author Contributions and Acknowledgments sections etc…
  • References section's style is not the one provided by the Microsoft Word template

Figure 3. There are no references to this figure in the text…

The Materials and Methods section can be improved better explaining the main references used for this review. For example, I suggest to re-organize lines 89-104 to better explain and emphasise the articles addressing the theme and the considered quantitative data.

The author writes at line 120-121 … “The magnitude of SOC erosion by fluvial processes varies widely (Table 1) depending on a range of factors (i.e., climate, soil, terrain, and land use)” … I suggest you to better explain this sentence adding some examples or references to better describe the role played by soil, terrain, and land use.

The Conclusion section is well supported by the results, but I suggest you re-write in order to better explain the relations between accelerated soil erosion and gaseous emissions.

Author Response

RESPONSE TO REVIEWERS’ COMMENTS ON THE ARTICLE: SOIL EROSION AND GASEOUS EMISSIONS

Both reviewers have made some pertinent comments. Therefore, the manuscript has been thoroughly revised with due consideration to each comment by both reviewers.

Specific revisions made are as follows:

Reviewer I

Thank you for excellent suggestions. Revisions made are outlined below:

  1. Yes, the Microsoft template is used to reorganize the manuscript layout
  2. The references now appear in order according to appearance in the paragraph, instead of the Table
  3. A section on Acknowledgement has been added
  4. The font is updated in the specific section to match the font in the rest of the manuscript.
  5. Figure 3 has been specifically cited (called out) in the text and appropriately discussed.
  6. The Material and Methods section has been expanded and improved as suggested
  7. The write up on the original lines 120-121 has been revised and specific references have been cited to document the impact of climate, soil, land use, and terrain. The corresponding references have been cited both from the table and the bibliography.

The revised version has been checked again for references, format template, and references

Reviewer 2 Report

The presented article has big significance as a role of eroded soil carbon in global atmosphere emission is discussed. Also author is focused attention to other gases, which can be formed in deposited soil material with large global warming potential in atmosphere. In form of minor corrections I suggest to mark activities which could to decrease volumes and rates of global soil erosion, and among of them – afforestation of agricultural lands in very useful soil practice of agroforestry management. Shelterbelts use atmospheric carbon and accumulate it in biomass during period of their growth. I recommend this article to publication.

Author Response

RESPONSE TO REVIEWERS’ COMMENTS ON THE ARTICLE: SOIL EROSION AND GASEOUS EMISSIONS

Both reviewers have made some pertinent comments. Therefore, the manuscript has been thoroughly revised with due consideration to each comment by both reviewers.

Specific revisions made are as follows:

Reviewer II

The second reviewer also provided very useful comments. Thus, the manuscript has been revised to incorporate all the suggestions as follows:

  1. Thank you for your comment “it has a big significance as a role of eroded SOC in the global atmosphere emission, and that the theme is focused on other GHGs which can be formed in the deposited material and which have a large GWP”. This comment is very meaningful.
  2. A new section has been included to outline examples of a few activities which could decrease volume and rates of erosion. Specific mention is made of the REDD, shelterbelt, cover cropping etc.
  3. Twelve new references have been cited in support of the section 2 above.

The revised version has been checked again for references, format template, and references